# Mg-incorporated sorbent for efficient removal of trace CO from H$_2$ gas

Gina Bang [1], Seongmin Jin [2], Hyokyung Kim[1], Kyung-Min Kim [3] ✉ & Chang-Ha Lee [1] ✉

Removal of trace CO impurities is an essential step in the utilization of Hydrogen as a clean energy source. While various solutions are currently employed to address this challenge, there is an urgent need to improve their efficiency. Here, we show that a bead-structured Mg, Cu, and Ce-based sorbent, Mg$_{13}$CuCeO$_x$, demonstrates superior removal capacity of trace CO from H$_2$ with high stability. The incorporation of Mg boosts sorption performance by enhancing the porous structure and Cu$^+$ surface area. Remarkably, compared to existing pelletized sorbents, Mg$_{13}$CuCeO$_x$ exhibits 15.5 to 50 times greater equilibrium capacity under pressures below 10 Pa CO and 31 times longer breakthrough time in removing 50 ppm CO in H$_2$. Energy-efficient oxidative regeneration using air at 120 °C allows its stable sorption performance over 20 cycles. Through in-situ DRIFTS analysis, we elucidate the reaction mechanism that Mg augments the surface OH groups, promoting the formation of bicarbonate and formate species. This study highlights the potential of MgCuCeO$_x$ sorbents in advancing the hydrogen economy by effectively removing trace CO from H$_2$.

Hydrogen (H$_2$) has emerged as a key player in the pursuit of clean energy and decarbonization, especially for industries in which the reduction of greenhouse gases is arduous and alternative solutions are limited. The International Energy Agency (IEA) reported that the annual H$_2$ demand reached 94 million metric tons (MtH$_2$ yr$^{-1}$) in 2021[1], representing a more than fivefold increase since 1975. Industries such as heavy transport, shipping, aviation, and heavy industry have become progressively reliant on H$_2$[2], while strategic frameworks developed by the European Commission[3] and the U.S. Department of Energy (DOE)[4] have accelerated the adoption of H$_2$ as an energy vector in various regions.

To ensure the successful utilization of H$_2$, particularly in proton exchange membrane fuel cells (PEMFCs), eliminating trace impurities such as carbon monoxide (CO) is of critical importance[5,6]. Because CO can bind to a platinum catalyst, it significantly reduces the ability of the catalyst to facilitate the target electrochemical reactions[7], and this deterioration is observed even at very low CO concentrations[8].

Therefore, the International Organization for Standardization (ISO) has introduced the ISO 14687:2019 standard, which limits the CO content in fuel-cell-grade H$_2$ to below 0.2 ppm[5]. Numerous H$_2$ purification methods have been proposed to comply with these stringent H$_2$ purity standards, including membrane separation[5,9], pressure (vacuum) swing adsorption (P(V)SA)[10,11], and preferential oxidation of CO (PROX)[12,13]. However, these technologies lead to a loss of H$_2$ even when targeting CO concentrations at a more lenient threshold than that imposed by ISO standards. CO can be successfully removed using more complex or combined separator systems, but this requires greater energy consumption and higher capital costs[14]. Moreover, the CO-PROX process, which continuously converts CO into carbon dioxide (CO$_2$), has two main concerns: (1) CO$_2$ decomposition can occur through in-situ reactions with H$_2$ (CO$_2$ + H$_2$ → H$_2$O + CO) or the electro-reduction of CO$_2$ (CO$_2$ + 2H$^+$ + 2e$^-$ → CO + H$_2$O) in PEMFCs[7], and (2) it produces CO$_2$, meaning that it fails to meet ISO standards requiring CO$_2$ levels lower than 2 ppm[15]. Therefore, the CO-PROX

[1]Department of Chemical and Biomolecular Engineering, Yonsei University, Seoul, Republic of Korea. [2]Institute of Chemical Sciences and Engineering, École Polytechnique Fédérale de Lausanne (EPFL), Lausanne, Switzerland. [3]Department of Biochemical Engineering, Gangneung-Wonju National University, Gangneung, Republic of Korea. ✉e-mail: kmkim@gwnu.ac.kr; leech@yonsei.ac.kr

process needs reaction energy and an additional separation unit for the removal of $CO_2$ and excess $O_2$ with additional regeneration energy. As a result, compliance with ISO standards using current $H_2$ purification methods is difficult due to the requirement of complex and energy-intensive processes.

To overcome these limitations, it is essential to develop sorbents with a robust CO sorption capacity (i.e., within the ppm range) that do not generate $CO_2$ and thus avoid the need for additional separation processes. However, the majority of CO sorbent research to date has concentrated on enhancing CO sorption within a pressure range of 100–200 kPa, which is more useful for CO production than for removal[16,17]. Additionally, $H_2$ products from $H_2$ production or purification processes contain a few percent or ppm of CO (Supplementary Table 1). Notably, while previous studies on powder sorbents have demonstrated a high CO sorption performance at 100–200 kPa of CO[16], a research gap exists regarding CO sorption capacity at ppm levels. Simultaneously, investigations into applying them in a stable pellet form for practical process applications are also lacking. The pelletization of powder sorbents often results in reduced sorption capacity and stability due to a decrease in surface area and porosity in pellets[18]. Given these considerations, the development of feasible materials that not only offer high sorption capacity at CO ppm levels but also maintain process stability in pelletized form is necessary for practical applications.

$Cu^+$ has been widely employed to facilitate CO sorption through both π-back donation and σ-bond formation[19], but sorbents containing $Cu^+$ experience aggregation during regeneration and oxidation when exposed to air[16,20]. In fact, a material suitable for CO sorption needs a high CO sorption capacity, oxidation resistance, and recycling ability, even at ultra-low CO concentrations. To address these challenges, a range of strategies have been employed to stabilize $Cu^+$ in sorbents and catalysts, with one approach involving Cu−Ce systems. These systems exploit the $Cu^+ + Cu^{2+} \leftrightarrow Ce^{4+} + Ce^{3+}$ redox cycle as an oxygen buffer, stabilizing $Cu^+$ ions[21]. Cu−Ce systems have exhibited great promise for CO removal applications, in particular as catalysts for low-pressure CO-PROX reactions[22].

It has been reported that incorporating magnesium into a Cu−Ce system promotes redox mechanisms, thus enhancing the water−gas shift reaction[5]. In particular, previous study has suggested that the introduction of Mg into Cu increases oxygen mobility[23], which can intensify the interaction between CO and the material surface[24]. Moreover, Mg-based materials with hierarchical structures have recently gained traction in enhancing sorption due to their high porosity[25,26], while also preventing the aggregation of metal alloys[27] and improving stability, strength[28], plasticity, and corrosion resistance[29,30]. Additionally, cyclic sorption stability has been achieved with the fabrication of a hierarchical spherical structure composed of Mg and Ce, further demonstrating the advantages of hierarchical configurations[31].

In this work, we synthesized a bead-structured Mg, Cu, and Ce-based sorbent (denoted as $CuCeO_x$ and $MgCuCeO_x$) that exhibited a significantly enhanced CO sorption performance at low pressure (i.e., <10 Pa). The role of Mg in $MgCuCeO_x$ was examined by adjusting the

Mg content and evaluating the resulting textural properties and morphological features of the sorbent. The physical properties of samples including Brunauer−Emmett−Teller (BET) surface areas were determined by the $N_2$ isotherms at 77 K. Since the total Cu surface area could be evaluated by the $N_2O$ chemisorption, X-ray photoelectron spectroscopy (XPS) and $N_2O$ chemisorption analysis were utilized to investigate the $Cu^+$ surface area (denoted as $S_{Cu+}$). CO sorption isotherms at 25 °C and breakthrough experiments using 50 ppm CO were conducted to assess the CO removal performance of $MgCuCeO_x$, which was compared with $CuCeO_x$ as a reference. Temperature-programmed desorption (TPD) experiments were carried out under oxygen or inert gas conditions to determine the influence of the regeneration gas, and the cyclic stability and working capacity of the sorbent for reactive CO removal under various regeneration conditions were investigated using thermogravimetric analysis (TGA). The CO sorption mechanisms of $MgCuCeO_x$ and $CuCeO_x$ with 50 ppm CO were determined using in-situ diffuse reflectance infrared Fourier transform spectroscopy (DRIFTS), which provided insights into the role of Mg in $MgCuCeO_x$ and identified the pathway for the surface reaction involving CO.

## Results

### Physicochemical properties of the Mg-promoted $CuCeO_x$ sorbent

$Mg_\alpha CuCeO_x$ beads and $CuCeO_x$ powders were successfully prepared by using a sol-gel combustion-assisted method. (Here, $\alpha$ represents the weight percentage of Mg relative to the total metal content.) Table 1 summarizes the textural properties of the $CuCeO_x$ and $MgCuCeO_x$ samples, as determined through the $N_2$ adsorption/desorption isotherms at 77 K depicted in Supplementary Fig. 2a. The isotherm curve for $CuCeO_x$ was classified as Type II using the BET isotherm classification, indicating a nonporous or macroporous material[32]. On the other hand, the isotherm curves for the $MgCuCeO_x$ samples were classified as Type IV with an H3 hysteresis loop, which is typical of a mesoporous material[33]. In Supplementary Fig. 2b, the pore size distribution determined using the Barrett−Joyner−Halenda (BJH) method revealed the development of mesopores with the addition of Mg. Notably, $Mg_{13}CuCeO_x$ had a narrow pore distribution centered around a diameter of approximately 2 nm. Table 1 shows that mesopores were more common than micropores in all of the samples. In addition, the surface area (52−232 $m^2 g^{-1}$), micropore volume (0.012−0.075 $cm^3 g^{-1}$), and mesopore volume (0.086−0.389 $cm^3 g^{-1}$) of the samples increased noticeably with higher levels of Mg. Thus, the addition of Mg led to the development of a Cu-Ce structure with numerous pores and a high surface area.

The scanning electron microscopy (SEM) image in Fig. 1a, b shows that $Mg_{13}CuCeO_x$ has a spherical bead morphology with a smooth surface, and the photograph of the bulk sorbents is presented in Supplementary Fig. 3. A spherical bead structure was observed for $MgCuCeO_x$ only when the Mg content was higher than 13 wt.%. Both Mg and Ce were essential to the formation of the spherical structure using sol-gel combustion as presented in literature[23,31], which results in powdered $CuCeO_x$. The element mapping results of the SEM image in

## Table 1 | Textural properties of the as-prepared sorbent samples

| Sorbents | BET surface area ($m^2 g^{-1}$) | Average pore diameter (nm) | Micropore volume[a] ($cm^3 g^{-1}$) | Mesopore volume[b] ($cm^3 g^{-1}$) |
|---|---|---|---|---|
| $CuCeO_x$ | 52 | 3.8 | 0.012 | 0.086 |
| $Mg_{13}CuCeO_x$ | 102 | 2.9 | 0.029 | 0.149 |
| $Mg_{23}CuCeO_x$ | 117 | 3.6 | 0.035 | 0.233 |
| $Mg_{52}CuCeO_x$ | 232 | 3.4 | 0.075 | 0.389 |

[a]by D-R method from 77 K $N_2$ isotherm.
[b]by BJH method from 77 K $N_2$ isotherm.

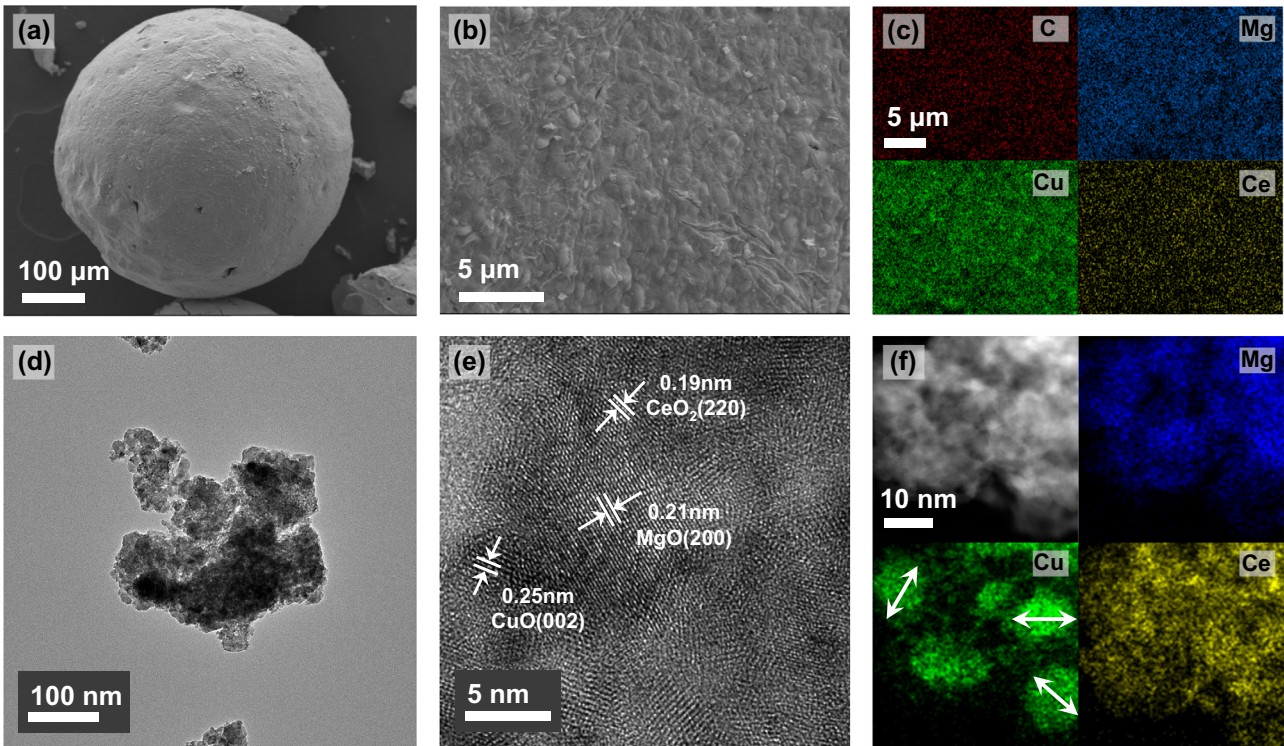

**Fig. 1 | Structural analysis of Mg$_{13}$CuCeO$_x$. a–c** SEM and EDS images, (**d**) TEM image of the ground sample, (**e**) HR-TEM image, and (**f**) STEM and corresponding elemental mapping images.

Fig. 1c confirmed the uniform distribution of Mg, Cu, and Ce, with a small amount of carbon from citric acid remaining even after calcination. The transmission electron microscopy (TEM) image in Fig. 1d shows a porous region and metal crystals in the Mg$_{13}$CuCeO$_x$ sample. MgO, CuO, and CeO$_2$ crystal planes were confirmed by measuring the interplanar spacing in the high-resolution transmission electron microscopy (HR-TEM) image (Fig. 1e).

Figure 2a depicts the DRIFTS spectra of as-prepared Mg$_{13}$CuCeO$_x$ and CuCeO$_x$ samples under He at 25 °C. Hydroxyl groups, bands within the range of 3300–3800 cm$^{-1}$, were observed and consisted of multi-coordinated surface OH group (bands <3650 cm$^{-1}$) and mono-coordinated surface OH groups (bands >3650 cm$^{-1}$)[34]. Mg$_{13}$CuCeO$_x$ had a larger amount of OH groups. The introduction of Mg led to a significant increase in defects, resulting in the formation of numerous surface groups, especially, peaks at 3652, 3578, and 3536 cm$^{-1}$[35].

X-ray diffraction (XRD) diffractograms of MgO, CuO, Cu$_2$O, and CeO$_2$ were used to analyze the crystal structure of the as-prepared MgCuCeO$_x$ and CuCeO$_x$ samples (Fig. 2b). The CuO and CeO$_2$ peaks became smaller and broader when Mg was introduced, indicating a reduction in the size of the crystallites. As shown in Fig. 2d, the as-prepared samples synthesized without a reduction step exhibited a reduction of Cu$^{2+}$ to Cu$^+$ in the range of 35–64%. The XRD patterns also revealed a relatively low intensity of Cu$_2$O peaks, suggesting a high dispersion of Cu$^+$ on the sorbent surface (Fig. 2b). The crystallite sizes of MgO, CuO, and CeO$_2$ were calculated from the XRD patterns (Table 2). The calculated crystal size corresponded to the width of the individual CuO shown in the TEM-EDS image in Fig. 1f. The CuO and CeO$_2$ crystals were both smaller in size within the MgCuCeO$_x$ samples compared with the CuCeO$_x$ samples. In particular, the size of the CeO$_2$ crystals decreased as the Mg content increased, which was consistent with the results of a previous study[35]. Both Mg and Ce are known to contribute to the reduction in the crystal size of CuO[23,35]. Because the MgCuCeO$_x$ samples were synthesized with a fixed Cu/Ce ratio in the

present study, an increase in the Mg content reduced the Ce content (Table 2). It was found that the optimal level of Mg that minimized the crystallite size of CuO (10.7 nm) was 13 wt.%.

The peaks in the temperature-programmed reduction using H$_2$ (H$_2$-TPR) results (Fig. 2c) were deconvoluted into highly dispersed CuO$_x$ species (peak α), CuO$_x$ species interacting with the support metals (peak β), and the bulk CuO$_x$ phase (peak γ)[36]. Bulk CuO$_x$ was predominantly observed in the CuCeO$_x$ sample (peak γ, 52%), while the MgCuCeO$_x$ samples consisted mainly of easily reducible Cu species (peaks α and β), with the largest area observed for Mg$_{13}$CuCeO$_x$.

The influence of the Mg content on $S_{Cu+}$ (Supplementary Equation (3)) and the chemical states of Cu and Ce species was investigated by combining XPS and N$_2$O chemisorption analysis (Fig. 2d, e, and Table 2). Because Cu$^+$ ions allow both π-complexation and σ-bonding for CO sorption[19], $S_{Cu+}$ can be considered an indicator of CO sorbent performance. $S_{Cu+}$ of CuCe-based sorbents is generally influenced by the Cu content, Cu dispersion, and Ce$^{3+}$ ratio. The presence of Mg enhanced the surface Cu ratio for CuCeO$_x$ (i.e., Cu dispersion, Table 2). In addition, the Ce$^{3+}$ ratio (Fig. 2e and Table 2) improved with an increase in the Mg content, which can slow down the oxidation of Cu$^+$ via the redox reaction Cu$^+$ + Ce$^{4+}$ ↔ Cu$^{2+}$ + Ce$^{3+}$. Because the samples were prepared using a fixed Cu/Ce ratio when adding Mg, an increase in the Mg content resulted in a decrease in the Cu content and the surface Cu$^+$ ratio in the MgCuCeO$_x$ samples (Fig. 2d and Table 2); despite this, the presence of Mg enhanced Cu dispersion and $S_{Cu+}$. In the present study, the highest $S_{Cu+}$ with the greatest Cu dispersion was observed for Mg$_{13}$CuCeO$_x$ (Table 2).

**CO sorption performance and dynamic behavior at trace levels**
CO sorption isotherms recorded at 25 °C for the samples are presented in Fig. 3a. The isotherm curves exhibited a steep increase in the low-pressure range (≤10 Pa). The results indicated that MgCuCeO$_x$ had a strong affinity for CO, which is an important

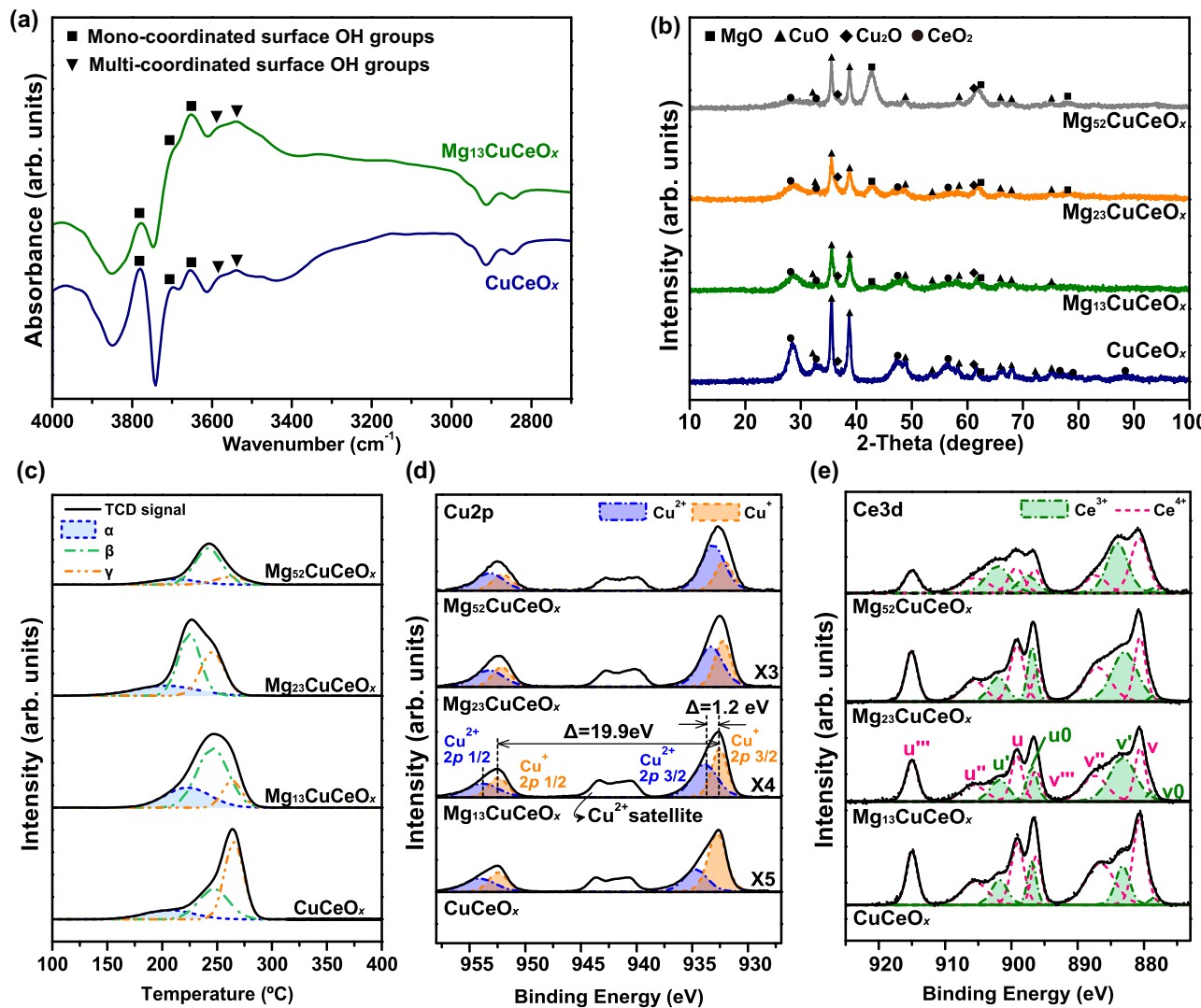

**Fig. 2 | Physicochemical analysis of the as-prepared sorbent samples. a** DRIFTS spectra; (**b**) XRD patterns; (**c**) H₂-TPR results: The ratio of the area of α, a highly dispersed CuOₓ species, to the combined area of (α + β + γ), signifying the total CuOₓ, is 16.6%, 20.3%, 27.5%, and 15.9%, from the topmost data downward, respectively; and XPS profiles: (**d**) Cu2*p* and (**e**) Ce3*d* spectra.

**Table 2 | Physicochemical properties of the as-prepared sorbent samples**

| Sorbents | Composition[a] (wt. %) | | | Crystallite size[b] (nm) | | | Peak area (%) | | | Cu dispersion (%)[e] | $S_{Cu^+}$ [a, d, e] $(m^2 g^{-1})$ |
|---|---|---|---|---|---|---|---|---|---|---|---|
| | **Mg** | **Cu** | **Ce** | **MgO** | **CuO** | **CeO₂** | $\frac{\alpha}{(\alpha+\beta+\gamma)}$ [c] | $\frac{Ce^{3+}}{(Ce^{4+}+Ce^{3+})}$ [d] | $\frac{Cu^+}{(Cu^{2+}+Cu^+)}$ [d] | | |
| CuCeOₓ | - | 57.8 | 28.1 | - | 16.7 | 4.5 | 15.9 | 23 | 64 | 3.7 | 9.2 |
| Mg₁₃CuCeOₓ | 10.6 | 45.5 | 22.0 | 4.2 | 10.7 | 2.6 | 27.5 | 40 | 46 | 14.7 | 21.0 |
| Mg₂₃CuCeOₓ | 16.5 | 35.9 | 17.3 | 4.9 | 11.1 | 2.3 | 20.3 | 37 | 44 | 13.4 | 14.2 |
| Mg₅₂CuCeOₓ | 33.3 | 20.2 | 9.6 | 4.7 | 13.0 | 0.9 | 16.6 | 43 | 35 | 11.0 | 5.2 |

[a]Estimated composition of Mg, Cu, and Ce ions by ICP-OES.
[b]Calculated from the XRD patterns by the Scherrer formula.
[c]Estimated from TPR results, and α indicates highly dispersed CuO species.
[d]Estimated from XPS results.
[e]Determined from N₂O chemisorption.

characteristic for the purification of H₂ to achieve fuel-cell-grade CO ppm levels. At 10 Pa (≈100 ppm CO at a total pressure of 100 kPa), the sorption capacity of the samples followed the order Mg₁₃CuCeOₓ > Mg₂₃CuCeOₓ > CuCeOₓ > Mg₅₂CuCeOₓ, which was consistent with $S_{Cu^+}$ (Table 2). However, at higher pressures (>90 Pa), Mg₅₂CuCeOₓ, which had the highest surface area (Table 1), exhibited higher sorption than CuCeOₓ, with a similar isotherm to Mg₂₃CuCeOₓ. Figure 3b depicts the relationship between $S_{Cu^+}$ and CO sorption capacity. At 10 Pa CO, the pressure range interested in this study, the sorption capacity displayed a positive correlation with increasing $S_{Cu^+}$. It can be deduced that $S_{Cu^+}$ is a critical determinant of sorption capacity under low CO concentration. Thus, our observations clearly indicated the necessity of well-dispersed Cu⁺.

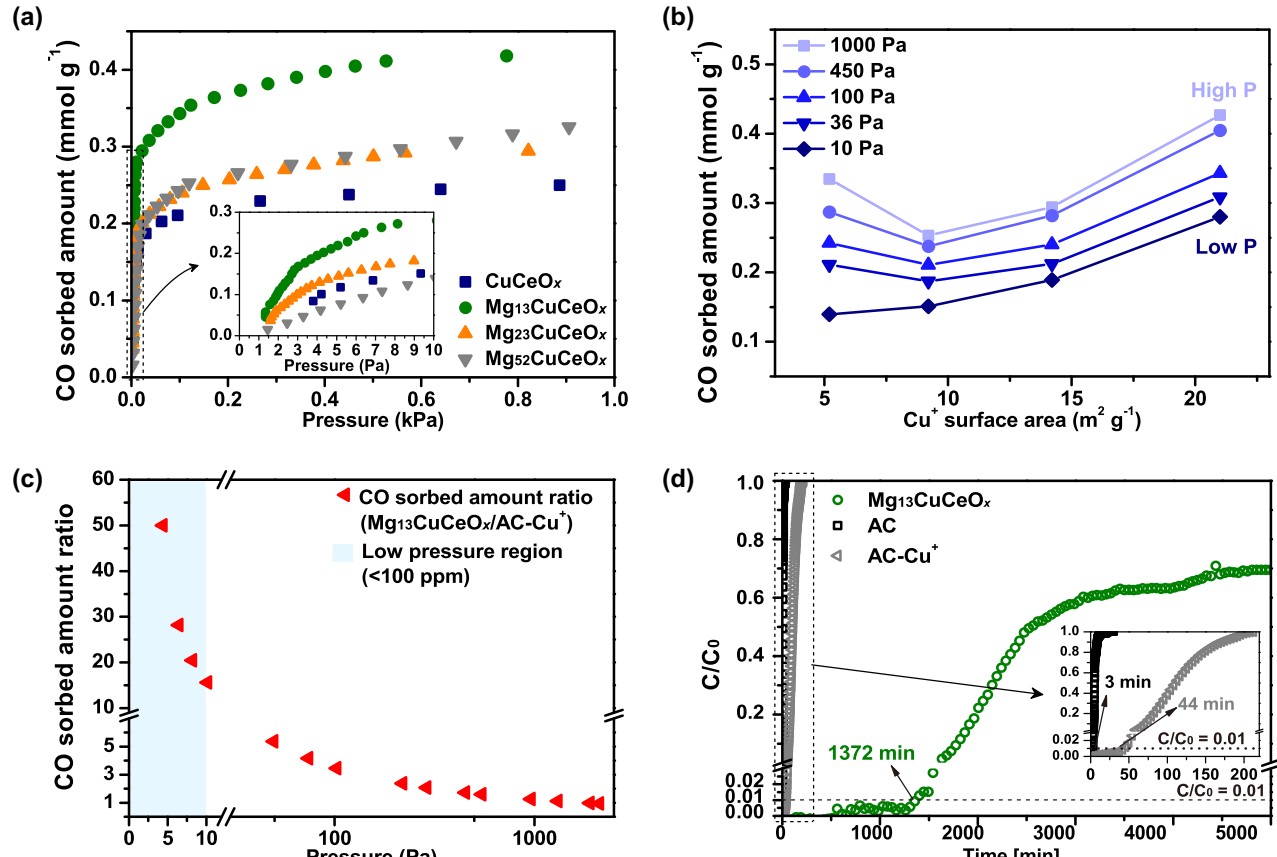

**Fig. 3 | Analysis of CO sorption performance. a** CO sorption isotherms for the as-prepared sorbent samples at 25 °C, (**b**) a comparison of CO sorbed amount of the sorbent as a function of Cu⁺ surface area at different pressures, (**c**) CO sorbed amount ratio between $Mg_{13}CuCeO_x$ and AC-Cu⁺, and (**d**) CO breakthrough curves at 25 °C under a flow of 50 ppm CO and $H_2$ balance (GHSV = 935 h⁻¹).

Numerous sorbents had reported considerable sorption performance, but powder sorbents often pose problems in packed or fluidized bed applications due to their associated high-pressure drop and potential for process contamination[16]. Thus, given these factors, a pellet Cu-impregnated activated carbon (AC-Cu⁺), with outstanding performance among the reported CO sorbents[37], served as a control group, and the performance evaluation incorporated both static and dynamic behavior. The CO sorption capacity of $Mg_{13}CuCeO_x$ was also compared with that of AC-Cu⁺ (Fig. 3c and Supplementary Fig. 4), sorbent with an excellent CO sorption capacity of 3.01 mmol g⁻¹ at 100 kPa[37]. A breakthrough experiment was conducted to evaluate the dynamic sorption performance of pristine activated carbon (AC), AC-Cu⁺, and $Mg_{13}CuCeO_x$ samples using 50 ppm CO ($H_2$ balance) as the feed gas (Fig. 3d). The breakthrough time was determined at $C/C_0 = 0.01$ (0.5 ppm CO), marking the red dotted line in the figure. Given the purified $H_2$ mix from 0 minutes to the breakthrough time, its overall CO content is less than 0.2 ppm, and the sorbent should be regenerated after the breakthrough time. The detection times of 0.5 ppm CO in AC and AC-Cu⁺ beds were specified in the insert figure, with the time for the $Mg_{13}CuCeO_x$ bed marked in green. The dynamic sorption capacity was calculated as the sorption amount up to the saturation time at $C = C_0$ ($S_{ads}$(ST)) or a long-term duration of 5750 min ($S_{ads}$(5750)), thereby providing comparative data for understanding sorption kinetic effects in practical applicability.

Compared to AC-Cu⁺, $Mg_{13}CuCeO_x$ exhibited 15.5 to 50 times higher sorption capacity at pressures below 10 Pa CO (Fig. 3c). As the pressure increased, this performance gap gradually diminished and ultimately inverted after 1700 Pa CO (Supplementary Fig. 4). These

results suggested that Mg and Ce enhanced the CO sorption performance of Cu in the ultra-low CO pressure region.

As presented in Fig. 3d, $Mg_{13}CuCeO_x$ exhibited the highest performance with a breakthrough time of 1372 min compared to 3 min for AC and 44 min for AC-Cu⁺, indicating 31 times longer in terms of the breakthrough time. In addition, $S_{ads}$(5750) of 0.209 mmol g⁻¹ was also approximately 30 times higher than $S_{ads}$(ST) of AC-Cu⁺. $S_{ads}$(5750) for $Mg_{13}CuCeO_x$ was 75% of the equilibrium sorption amount, although the contact time between the sample and CO was limited during dynamic operation. Given that $S_{ads}$(ST) of AC-Cu⁺ was only 39% of the equilibrium capacity, $Mg_{13}CuCeO_x$ demonstrated a relatively rapid sorption rate during the initial breakthrough period. A concentration plateau was observed in the breakthrough curve for $Mg_{13}CuCeO_x$ due to chemisorption, while it steadily sorbed a certain amount of CO even after 5750 min. Consequently, $Mg_{13}CuCeO_x$ has the potential to be successfully employed for the removal of low concentrations of CO even under dynamic conditions.

## Stability of cyclic CO sorption using oxidative regeneration
The effective regeneration of a sorbent is a key factor in practical applications because this process typically requires significant energy and capital resources. In order to determine the regeneration conditions required for the repeated use of $Mg_{13}CuCeO_x$, its desorption properties under an inert gas (He) were assessed using CO-TPD analysis (Fig. 4a). The CO species chemisorbed onto the sorbent surface were mainly desorbed as $CO_2$ at >100 °C, with a small quantity of CO released between 100 °C and 400 °C. It indicated that the strongly chemisorbed CO begins to be desorbed in the form of $CO_2$ at elevated temperatures, instead of ambient temperatures. This phenomenon

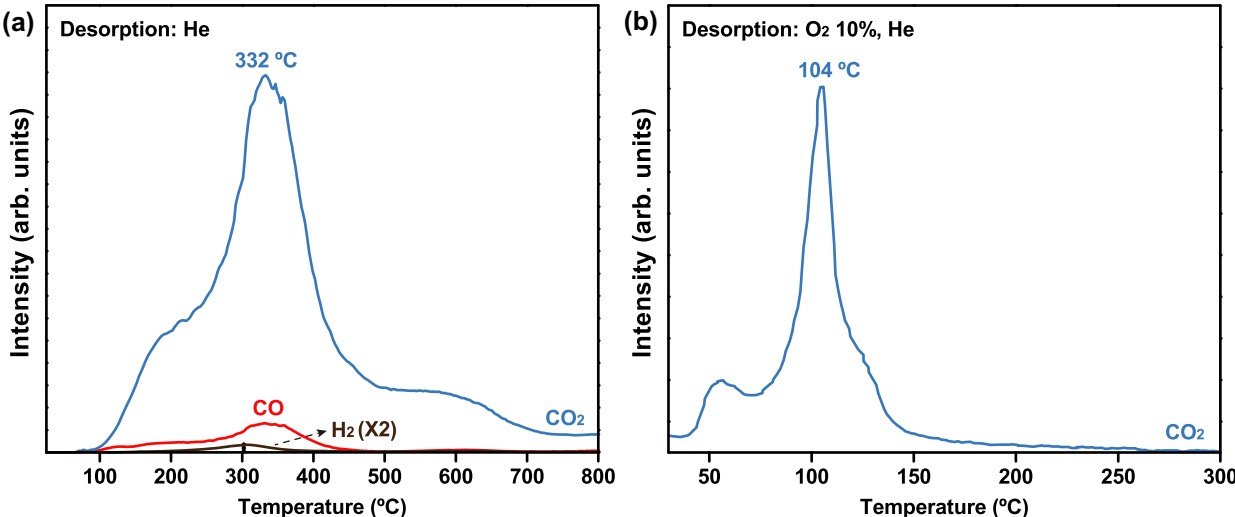

**Fig. 4 | CO-TPD profiles for Mg₁₃CuCeOₓ.** CO-TPD profiles under (**a**) He and (**b**) O₂ 10% (He balance).

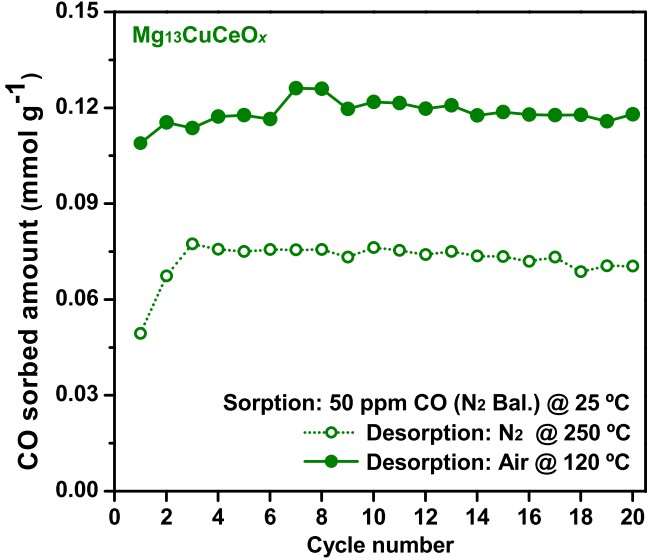

**Fig. 5 | Cyclic sorption-regeneration tests for Mg₁₃CuCeOₓ.** Cyclic tests with sorption for 60 min at 25 °C under 50 ppm CO (N₂ balance) and regeneration for 30 min in air (21% O₂ and N₂ balance) at 120 °C or in N₂ at 250 °C.

was corroborated by an additional breakthrough test (Supplementary Fig. 5). At 25 °C, any $CO_2$ molecules were not observed until the breakthrough curve reached the feed concentration, further emphasizing that CO was being chemisorbed without concurrent emission of $CO_2$. Furthermore, the absence of $CO_2$ at ppb levels during the breakthrough tests indicated that the $MgCuCeO_x$ sorbent can satisfy the fuel-cell grade hydrogen quality with a successful trace CO and without $CO_2$ generation during a $H_2$ purification step. Consequently, the $MgCuCeO_x$ sorbent functioned as a $CO_2$-free CO sorbent, rather than a catalyst for CO conversion, during CO removal step at ambient temperatures.

Trace $H_2$ was detected at around 330 °C, which was associated with the interaction of CO with the OH groups on the sorbent surface (Fig. 2a). $CO_2$ peaks were observed at high temperatures (200, 330, and 580 °C), representing a similar trend to that observed for the α, β, and γ peaks of $Mg_{13}CuCeO_x$ in the TPR analysis (Fig. 2c). The $CO_2$ peaks at these temperatures originated from the desorption of CO species from the highly dispersed $CuO_x$, $CuO_x$ species interacting with the support metals, and the bulk $CuO_x$ phase, respectively. The observed

desorption behavior can be ascribed to either the disproportionation reaction ($2CO \rightarrow CO_2 + C$)[38] or desorption with oxygen on the surface of a metal oxide ($CO + O_{sorbent} \rightarrow CO_2$)[39], with both pathways potentially leading to the deterioration of the sorbent. In order to address this issue, the present study employed oxidative regeneration on the CO sorbent. Utilizing oxygen as the regeneration gas, degradation reactions can be suppressed[40,41]. It should be noted that the energy requirement for the $CO + O_{regeneration} \rightarrow CO_2$ reaction is considerably lower than that for the aforementioned degradation reactions.

Figure 4b presents the CO-TPD results when oxygen (O₂ 10%, He balance) was employed as the regeneration gas. Using oxidative regeneration, the temperature required for maximum desorption was significantly lower at 104 °C compared with 332 °C under inert gas flow. Additionally, desorption occurred only in the form of $CO_2$. Consequently, employing oxidative regeneration represented an effective energy-efficient solution to sorbent deterioration.

Figure 5 presents the results of the stability test of $Mg_{13}CuCeO_x$ over 20 cycles following 30-min pretreatment with air or $N_2$ gas at 450 °C. During the sorption process, 50 ppm CO ($N_2$ balance) was employed, while air or $N_2$ was used as the desorption medium. The sorption and regeneration periods were 1 h and 30 min, respectively. The samples pretreated and regenerated using air exhibited a cyclic working capacity of 0.12 mmol g⁻¹, efficiently sorbing most of the CO present (>99%) with excellent stability. These findings indicated that the $Cu^+$ ions were conserved after oxidative regeneration, and this was corroborated by the XPS (Fig. 2d) and $S_{Cu^+}$ results (Table 2) obtained immediately after sorbent synthesis using air calcination.

In contrast, the sample pretreated with $N_2$ experienced a decline in its sorption capacity of approximately 50% compared with air pretreatment, and a progressive reduction in its sorption capacity was observed with each successive sorption–desorption cycle, which was attributed to a lower sorption rate. This hypothesis is supported by previous studies reporting the impact of $O_2$ pre-adsorption on the sorption–oxidation rate of CO[42,43]. Overall, even when air, which is significantly more cost-effective and readily available than reducing or inert gases, was utilized for pretreatment or regeneration, the $Cu^+$ ions on the $MgCuCeO_x$ surface remained intact and the sorption rate remained high.

## Mechanistic study of CO chemisorption

CO DRIFTS spectra of $CuCeO_x$ and $Mg_{13}CuCeO_x$ under 50 ppm CO and 1% CO were compared with gaseous CO peaks of inert KBr (without a sorbent sample) under 1% CO (Fig. 6a). Under 1% CO, both $Mg_{13}CuCeO_x$ and $CuCeO_x$ exhibited peaks at 2174 and 2108 cm⁻¹, associated with

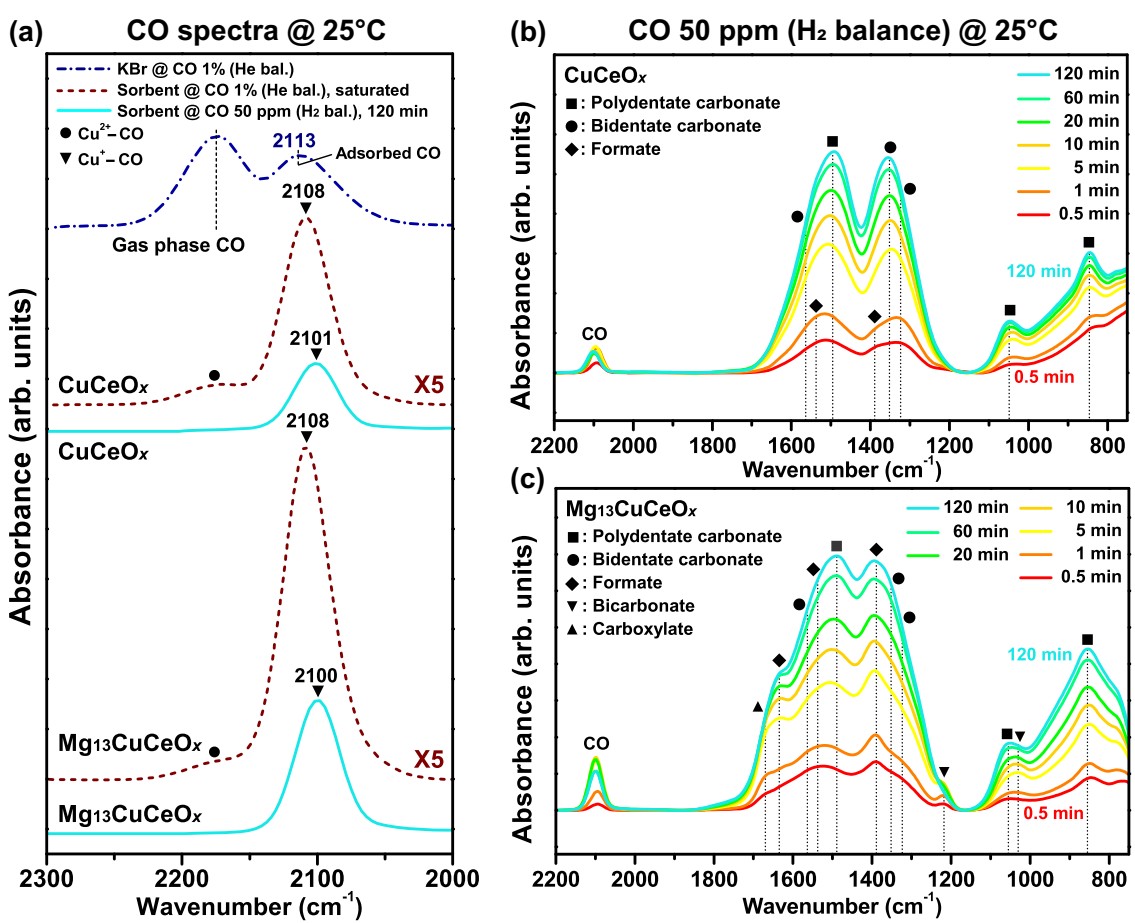

**Fig. 6 | DRIFTS Spectra Comparison. (a)** DRIFTS spectra comparison. (**a**) DRIFTS spectra of CO sorbed on $Mg_{13}CuCeO_x$ and $CuCeO_x$ at different CO compositions and In-situ DRIFTS spectra for (**b**) $CuCeO_x$ and (**c**) $Mg_{13}CuCeO_x$ at 25 °C in 50 ppm CO ($H_2$ balance).

$Cu^{2+}$−CO and $Cu^+$−CO, respectively[44]. As shown in the magnified view of Fig. 6a in Supplementary Fig. 7, the band intensity ratio of the $Cu^+$−CO to $Cu^{2+}$−CO band was larger in $Mg_{13}CuCeO_x$. This indicated that Mg increases the surface area of $Cu^+$, as shown in Table 2, enhancing its accessibility to CO (Fig. 3). Furthermore, the bands, corresponding to $Cu^+$−CO of $CuCeO_x$ and $Mg_{13}CuCeO_x$, exhibited a red shift compared to gaseous CO, and this shift was larger in 50 ppm CO case than in 1% CO case. The band shift for $Cu^+$−CO can result from the contribution of σ and π bonds. The red shifts implied the dominance of π bonding during CO sorption onto the samples[20], which was amplified at lower pressures.

In-situ DRIFTS was employed to investigate the CO sorption mechanisms at extremely low pressures and to elucidate the role of Mg in the sorbent. In-situ DRIFTS spectra were recorded for the $Mg_{13}CuCeO_x$ and $CuCeO_x$ samples during the breakthrough experiment using 50 ppm CO gas ($H_2$ or $N_2$ balance) at 25 °C (Fig. 6b, c and Supplementary Fig. 6). The peak assignments and structure of the analogous carbonates produced in the present study are presented in Supplementary Table 2. For $CuCeO_x$ (Fig. 6b), polydentate carbonate (1492, 1052, and 850 $cm^{-1}$, indicated by the square symbol) and bidentate carbonate (1563, 1350, and 1323 $cm^{-1}$, indicated by the circle symbol) were the predominant sorbed CO species for both $H_2$ and $N_2$ as the balance gas. These species became dominant after 5 min, which was consistent with the carbonate species that formed during chemisorption on the surface of a CuO-based commercial catalyst[45]. Minor contribution of CO physisorption was evidenced by the CO peak at 2100 $cm^{-1}$, which was saturated after about 60 min. Additionally, trace peaks for formate species (1392 and 1535 $cm^{-1}$, indicated by the diamond symbol) were observed within 1 min.

During CO chemisorption on $Mg_{13}CuCeO_x$ (Fig. 6c), polydentate carbonate and bidentate carbonate were observed as the main peaks, and other routes were identified, which could be attributed to the presence of Mg. Noticeably, formate species, which were indicated by the peaks at 1635, 1535, and 1392 $cm^{-1}$ ($v_{as}(COO)$[46], $v_s(OCO)/v_{as}(OCO)$[47], and C−H bending[48], respectively), were a key pathway. A carboxylate peak at 1670 $cm^{-1}$ and bicarbonate peaks at 1218 and 1038 $cm^{-1}$ were also observed as secondary peaks. The bicarbonate peak at 1218 $cm^{-1}$, corresponding to $v(CO_3)$, increased until 10 min and then decreased, indicating that bicarbonate acts as an intermediate during the chemisorption process. Interestingly, the formation and disappearance of bicarbonate occurred more slowly when $N_2$ was used as the balance gas (Supplementary Fig. 6b), suggesting that the presence of $H_2$ in the feed can accelerate the bicarbonate-related chemisorption process. The observation that bicarbonate ($HCO_3^-$) and formate ($HCO_2^-$) species were exclusively formed on $Mg_{13}CuCeO_x$ even in the presence of the $N_2$ balance gas suggested that the OH groups on the sorbent surface played a significant role in the formation of these species. The chemisorption mechanisms for $Mg_{13}CuCeO_x$ were thus assumed to involve an additional pathway associated with the surface OH groups, leading to the rapid formation of bicarbonate ($HCO_3^-$) species during the initial stages of the chemisorption process, followed by their conversion into formate ($HCO_2^-$).

Based on the spectral analysis, a possible catalytic reaction pathway for CO on the surface of $MgCuCeO_x$, was proposed (Fig. 7). In Step I (Fig. 7a−c), a CO molecule interacts with an OH group on the sorbent surface. This interaction is facilitated by the weakly basic OH groups, which promote the generation of bicarbonate species[49]. When CO is introduced to the $MgCuCeO_x$ surface, it acquires an additional oxygen

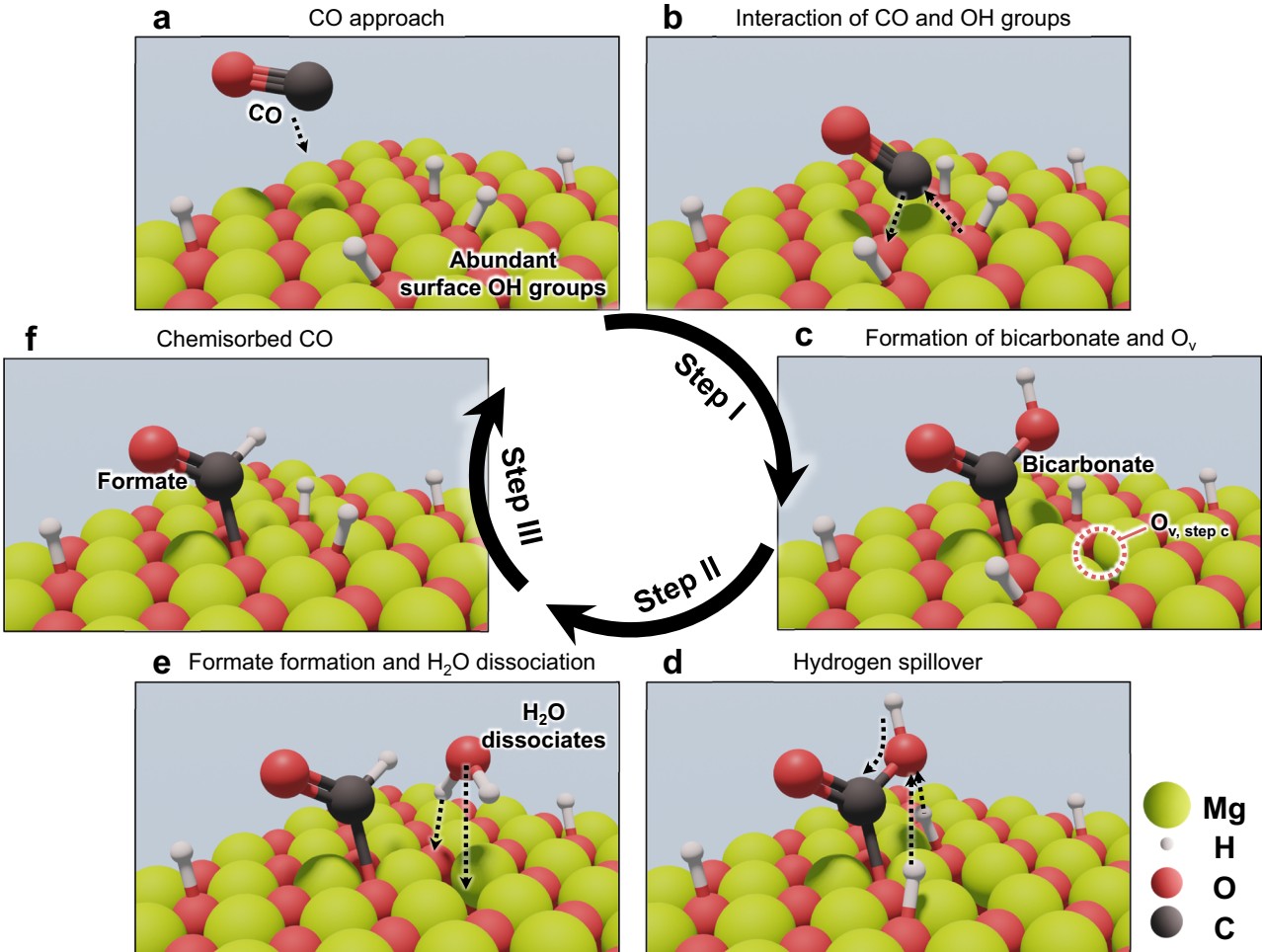

**Fig. 7 | Schematic diagram of the OH group-based CO chemisorption pathway on MgCuCeO$_x$.** (**a**) CO approach, (**b**) interaction of CO and OH groups, (**c**) formation of bicarbonate and oxygen vacancy (O$_v$), (**d**) hydrogen spillover, (**e**) formate formation and water dissociation, and (**f**) chemisorbed CO.

atom from the surface, leading to the creation of an oxygen vacancy (O$_{v, \text{step c}}$) and bicarbonate. In Step II (Fig. 7d, e), the bicarbonate species reacts with H atoms from neighboring surface OH groups to generate water and formate, as reported by previous studies[50]. In Step III (Fig. 7f), the water dissociates into OH and H at the O$_{v, \text{step c}}$, and surface oxygen, respectively[34], thus regenerating OH groups. The presence of Mg in the sorbent contributes to the formation of surface OH groups (Fig. 2a), playing a crucial role in this CO chemisorption pathway and enhancing the sorption capacity of the sorbent. Consequently, Mg$_{13}$CuCeO$_x$ exhibited high CO removal performance with excellent stability in the cyclic test (Fig. 5).

## Discussion

We successfully developed a highly stable bead-structured Mg, Cu, and Ce-based sorbent for the effective removal of trace CO from H$_2$ gas. By including Mg in the CuCeO$_x$-based sorbent, we achieved a pore-rich bead structure and a large surface area. The optimal Mg ratio was found to enhance the dispersion by producing smaller Cu crystals, which was essential to the superior sorption performance. MgCuCeO$_x$ retained a high Cu$^+$ surface area even after air calcination, which was attributed to the presence of Mg increasing the proportion of Ce$^{3+}$ to promote the redox reactions. Mg$_{13}$CuCeO$_x$ exhibited superior CO sorption at both equilibrium and under dynamic conditions at ultra-low pressures below 10 Pa (≈100 ppm CO at a total pressure of 100 kPa), a performance that surpassed that of previously reported pelletized sorbents. This suggests that Mg$_{13}$CuCeO$_x$ can be efficiently employed as a sorbent for the production of fuel-cell-grade H$_2$.

Moreover, we demonstrated oxidative regeneration as a viable strategy for the repeated use of a sorbent. The pre-adsorption of oxygen had a beneficial influence on the reaction rate during the CO sorption–oxidation process for MgCuCeO$_x$. This ensured that Mg$_{13}$CuCeO$_x$ maintained its sorption capacity during cyclic tests, highlighting its stability and the effectiveness of air for pretreatment and regeneration. We also showed that oxidative regeneration is an energy-efficient method without any sorbent degradation.

We explored the CO chemisorption mechanisms associated with MgCuCeO$_x$ using in-situ DRIFTS, highlighting the critical role of Mg in enhancing the sorption capacity. Mg increased the abundance of surface OH groups, which became active centers in the catalytic reactions for the formation of bicarbonate and formate species. This process ensured continuous CO removal and demonstrated the importance of incorporating Mg into CuCeO$_x$-based sorbents. Therefore, in the breakthrough test, Mg$_{13}$CuCeO$_x$ continuously removed a certain amount of CO even after 5000 min of exposure to a 50 ppm CO stream at a GHSV of 935 h$^{-1}$.

Consequently, the presence of Mg contributed to improving the structural and chemical properties of the sorbent, thus enhancing its sorption capacity and cyclic stability. Therefore, MgCuCeO$_x$ represents a promising, effective, and stable sorbent for ultra-low levels of CO, one that does not discharge CO$_2$ during the removal process. Since MgCuCeO$_x$ proved a high removal potential of trace CO in H$_2$ and was readily regenerated at 120°C under an air flow, simple systems using MgCuCeO$_x$ can cost-effectively remove trace CO. These results highlight the potential of this sorbent for practical

applications targeting the generation of ultra-high-purity $H_2$, especially for use in fuel cells.

## Methods

### General

All reagents were purchased from vendors and used without further purification. The structural properties were acquired from $N_2$ adsorption/desorption isotherms at 77 K with an Autosorb IQ instrument (Quantachrome, version 5.21). The morphology and elemental distribution were examined by a combination of FE-SEM, TEM, and EDS techniques using JSM-7610F-Plus (JEOL, Ltd.), JEM-ARM 200 F (NEOARM), (JEOL, Ltd.), and X-MAX TSR (OXFORD Instruments), respectively. X-ray diffraction (XRD) data was collected from an Ultima IV diffractometer (Rigaku) with Cu-Kα radiation (λ = 1.54 Å). The $H_2$-TPR measurement was carried out using a ChemBET Pulsar TPR/TPD unit (Quantachrome). XPS analyses were performed using Thermo Fisher Scientific with monochromated Al Kα radiation as the excitation source. Cu dispersion and the $S_{Cu}$ were determined via selective $N_2O$ chemisorption experiments conducted at 50 °C.

### Sorbent synthesis

$MgCuCeO_x$ beads and $CuCeO_x$ powders were synthesized using a sol-gel combustion-assisted method. Magnesium nitrate hexahydrate (0.01 mol) with the desired amount of copper nitrate trihydrate, cerium nitrate hexahydrate, and citric acid monohydrate was dissolved in deionized water. The detailed ratios of each substance are described in the Supplementary Methods. The solution was then stirred in an oil bath at 80 °C for 5 h and dried at 90 °C for 2 h. The dried sample was ground and sieved (mesh size of 250–600 μm), followed by drying overnight at 110 °C. Lastly, the resulting sample was calcined under a flow of 21% $O_2$ (with an $N_2$ balance, hereafter referred to as air for simplicity) by ramping the temperature to 450 °C at a heating rate of 1 °C min⁻¹ and maintained for 10 h.

AC-Cu⁺ was synthesized using the impregnation method[37]. Pristine AC support was pre-treated at 700 °C for 3 h under $H_2$ flow. The pre-treated AC was then impregnated with 4.26 mmol of copper per gram of sorbent by stirring for 1 h at 60 °C in a solution of copper formate tetrahydrate and HCl. The impregnated sorbent was then washed, filtered, and dried at 60 °C overnight. The final AC-Cu⁺ was obtained by activating the dried sorbent under $N_2$ flow at 100 mL min⁻¹ and 300 °C for 3 h. The details of these experiments can be found in Supplementary Methods.

### CO sorption and desorption test

CO sorption isotherms were acquired using a commercial sorption analyzer (Autosorb IQ, Quantachrome, version 5.21) and the conventional static volumetric method under pressures of up to 1 kPa. Before analysis, the $CuCeO_x$ and $MgCuCeO_x$ samples underwent vacuum degassing at 280 °C for 8 h to eliminate adsorbed impurities.

Prior to the breakthrough experiment, AC was treated at 120 °C for over 12 h under He flow, AC-Cu⁺ was treated at 200 °C for 3 h, and $Mg_{13}CuCeO_x$ was treated with air at 280 °C for 30 min. The breakthrough apparatus (Supplementary Fig. 1) with a CO-infrared analyzer with a detection limit of 0.02 ppm (CO-IR, Everise) was utilized to perform breakthrough tests. All experiments were conducted at 1 bar, 25 °C, and a GHSV of 935 h⁻¹.

The desorption behavior of $Mg_{13}CuCeO_x$ was examined using CO-TPD with a Chem BET Pulsar TPR/TPD unit in conjunction with an online mass spectrometer (MS; HPR 20, Hiden Analytical Ltd.). Following chemisorbent pretreatment under the same conditions as for the breakthrough experiment, the temperature was reduced to 25 °C and the samples were purged with He. The samples were then exposed to 1% CO (He balance) at a flow rate of 50 ml min⁻¹ at 25 °C for 1 h to reach saturation. A He purge was then conducted until no residual gas was detected by the MS. The regeneration of the chemisorbent

samples was then conducted using either He or air under heating up to 800 °C.

The cyclic CO uptake of $Mg_{13}CuCeO_x$ was investigated using TGA (TGA 4000, Perkin Elmer) at 1 bar. Prior to each experiment, the $Mg_{13}CuCeO_x$ sample was subjected to pretreatment at 450 °C for 30 min under air or $N_2$ (40 mL min⁻¹). Cyclic sorption was conducted with the removal step occurring under a flow of 50 ppm CO ($N_2$ balance, 40 mL min⁻¹) at 25 °C for 60 min and the regeneration step under air at 120 °C or $N_2$ at 250 °C (40 mL min⁻¹) for 30 min. The details of these experiments can be found in Supplementary Methods.

### In-situ DRIFTS experiments

In-situ IR spectra were collected using a Nicolet iS10 FT-IR instrument equipped with a DRIFT accessory and a cell from PIKE Technologies. The spectra were recorded with a resolution of 4 cm⁻¹ and 64 scans, and an MCT cryodetector was used for signal detection. The samples were pretreated under air with a flow rate of 50 mL min⁻¹ at 450 °C for 30 min using a high-temperature cell equipped with Zn-Se windows. The background IR spectrum was measured at the sorption temperature of 25 °C before exposing the pretreated samples to 50 ppm CO ($H_2$ or $N_2$ balance) in a feed stream with a flow rate of 50 mL min⁻¹ at 25 °C for 120 min.

## Data availability

The data to support the findings of this study are available from the corresponding authors upon request.

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

## Acknowledgements

This work was supported by the National Research Foundation of Korea (NRF), funded by the Ministry of Science and ICT (2019K1A4A7A03113187).

## Author contributions

C.-H.L. designed the direction of research. G.B., S.J., and C.-H.L. conceptualized the synthetic experiments. G.B. executed and interpreted

the synthetic, low-pressure sorption, column breakthrough tests, and DRIFTS experiments. H.K. analyzed the XPS and crystallographic data. G.B. and S.J. contributed to the DRIFTS data analysis. K.-M.K. and C.-H.L. provided discussions and validated the data and analyses. G.B., K.-M.K., and C.-H.L. wrote the manuscript. All authors reviewed and contributed to the final manuscript.

## Competing interests

The authors declare no competing interests.
