## [Peer Review File · Nature Communications]

Mg-incorporated sorbent for efficient removal of trace CO from H₂ gasReviewer #1 (Remarks to the Author):

The authors wish to improve CO sorption for hydrogen purification by incorporating magnesium into the Cu-Ce systems. As it declared in the introduction section, both CO and CO₂ with even trace amount are harmful to the PEMFC. It seems from DRIFTS and the proposed chemisorption mechanism that CO₂ would form and escape into stream. It is also needed to evaluate the concentration of CO and CO₂ in the effluent H₂ stream and show whether it can reach the criteria mentioned in the introductory section (e.g., ISO 14687:2019). This work is premature to be considered for publication.

Reviewer #3 (Remarks to the Author):

In this study, bead-structured Mg, Cu, and Ce-based sorbent (Mg₁₃CuCeO_x) was synthesized for the removal of CO in hydrogen rich syngas for the purpose of using as fuel for hydrogen fuel cell. This is a very important step for the implementation of hydrogen fuel cell. It has the potential to be published in NC, but the current version is still far from my expectation of an NC paper. Some important points should be addressed in the next revision.

1. "Furthermore, powder-type sorbents have been reported to have a high CO sorption equilibrium at certain CO concentrations" I do not agree with this statement. The CO sorption equilibrium is a thermodynamic concept, so it should not be affected by the physical shape. The authors need to provide more explanations on how physical shape affect the equilibrium.
2. The above sentence should have a comma instead of a full stop at the end.
3. "X-ray photoelectron spectroscopy (XPS) and N₂O chemisorption analysis were utilized to investigate the Cu + surface area (denoted as S_{Cu+})" I think it is N₂ instead of N₂O. Another point is why Cu⁺ surface area is measured. Is not it the total surface area of the material? Does Cu⁺ have surface area?
4. The experiment was conducted in 50ppm CO environment. However, most syngas has 10-50% CO. I would like to see the authors' material performance for high CO concentration. The integration of area below CO in Fig. 3d indicated it has pretty large CO capacity. So it might be possible to apply the material in high CO concentration gas.
5. Fig. 1a showed a bead. I would like to see a larger range of the picture. Is everywhere the material ball shape? Or only some of them are in ball shape?
6. The caption of Fig. 2 has typo. Fig. 2e's caption is missing.
7. The reason to choose Mg is not clearly demonstrated. Please elaborate the reason why Mg should be used in this study.
8. The authors criticized CO-PROX technology due to the generation of CO₂. What if the system loads some CO₂ sorbents? Is the problem possibly to be resolved?
9. PROX's full name is not provided in this article.

Responses and revisions to reviewers' comments

Manuscript title: Mg-promoted sorbent and oxidative regeneration amplify trace CO removal efficiency from hydrogen

Manuscript Number: NCOMMS-23-24617

***Reviewer #1:** The authors wish to improve CO sorption for hydrogen purification by incorporating magnesium into the Cu-Ce systems. As it declared in the introduction section, both CO and CO₂ with even trace amount are harmful to the PEMFC. It seems from DRIFTS and the proposed chemisorption mechanism that CO₂ would form and escape into stream. It is also needed to evaluate the concentration of CO and CO₂ in the effluent H₂ stream and show whether it can reach the criteria mentioned in the introductory section (e.g., ISO 14687:2019). This work is premature to be considered for publication.*

Response: Thank you for your insightful comments, particularly highlighting the importance of assessing the potential release of CO₂ during CO sorption.

As emphasized in the experimental, the CO removal experiment was conducted at 25°C without any supplement of O₂. CO was chemisorbed in various carbonate forms as shown in *in situ* DRIFTS (refer to Fig. 6), but the chemisorbed CO was desorbed as CO₂ at >100°C (as seen in CO-TPR in Fig. 4). Furthermore, although not as strong as CO, MgCuCeO_x exhibited substantial CO₂ sorption capacity at low concentrations as shown in the figure below (It was not added to the manuscript because of out of scope in the study). Therefore, even under unlikely events of CO₂ generation, it wouldn't escape the sorption bed during CO removal step at ambient temperature.

To further confirm the chemisorption of CO, we adapted the experimental setup to detect CO₂ accurately in the effluent stream during a CO breakthrough test. We configured a new analysis system consisting of a mass spectrometer along with a GC equipped with an FID and a methanizer, which enables CO₂ detection under <ppm levels. Then, a new breakthrough test was conducted at 25°C until the breakthrough curve reached the feed concentration (50 ppm CO in H₂ balance).

In the result, any CO₂ was not detected in the effluent gas through the breakthrough test, substantiating that MgCuCeO_x operates as a CO₂-free CO sorbent and not as a catalyst for

CO oxidation at 25°C. Therefore, we confidently confirmed that the proposed MgCuCeO_x sorbent could be used to simultaneously comply with the CO concentration criterion specified by ISO 14687:2019 without the issue of CO_2 generation during a CO removal step. We revised the manuscript to present the results clearly with the addition of the new experimental breakthrough test as the supplementary data.

Figure. CO_2 sorption isotherm for the as-prepared $\text{Mg}_{13}\text{CuCeO}_x$ at 25°C.

Page 12, lines 234–249 (Added)

The CO species chemisorbed onto the sorbent surface were mainly desorbed as CO_2 at $>100^\circ\text{C}$, with a small quantity of CO released between 100°C and 400°C . It indicated that the strongly chemisorbed CO begins to be desorbed in the form of CO_2 at elevated temperatures, instead of ambient temperatures. This phenomenon was corroborated by an additional breakthrough test (Fig. S5). At 25°C , any CO_2 molecules were not observed until the breakthrough curve reached the feed concentration, further emphasizing that CO was being chemisorbed without concurrent emission of CO_2 . Furthermore, the absence of CO_2 at ppb levels during the breakthrough tests indicated that the MgCuCeO_x sorbent can satisfy the fuel-cell grade hydrogen quality with a successful trace CO and without CO_2 generation during a H_2 purification step. Consequently, the MgCuCeO_x sorbent functioned as a CO_2 -free CO

sorbent, rather than a catalyst for CO conversion, during CO removal step at ambient temperatures.

Trace H₂ was detected at around 330°C, which was associated with the interaction of CO with the OH groups on the sorbent surface (Fig. 2a). CO₂ peaks were observed at high temperatures (200, 330, and 580°C), representing a similar trend to that observed for the α, β, and γ peaks of Mg₁₃CuCeO_x in the TPR analysis (Fig. 2c).

Fig. S1 (Revised), Supplementary data Page 8 (Added)

Fig. S1. Schematic diagram of breakthrough experimental apparatus.

In the long-term breakthrough experiment presented in Fig. 3d, we employed a monitoring approach illustrated by the blue line in Fig. S1, which exclusively focused on CO detection via infrared (IR) spectroscopy. Conversely, the breakthrough experiment presented in Fig. S5 was designed for the simultaneous detection of CO and CO₂ at ppb levels until the breakthrough curve reached the feed concentration. For this full breakthrough curve experiment, we used a more complex analysis setup that included a gas chromatograph (GC) equipped with a flame ionization detector (FID) and a methanizer, along with a mass spectrometer (MS). This experimental methodology is represented by the red line in Fig. S1.

Fig. S5 (Added)

Fig. S5. CO breakthrough curve at 25°C under a flow of 50 ppm CO and H₂ balance (GHSV = 4185 h⁻¹).

The breakthrough test was conducted under a harsher flow condition than the long-term breakthrough experiment displayed in Fig. 3d. The analysis system showed a noise level signal from the feed gas because of the CO₂ impurity in the feed gas. Then, through the breakthrough experiment, the noise level CO₂ signal did not change. The results revealed no detection of CO₂ even at CO₂ at ppb levels, and the breakthrough curve could reach 50 ppm of CO (the feed concentration). Considering the analysis system, which could detect CO₂ at ppb level, the MgCuCeO_x sorbent satisfied the removal of trace CO in H₂ for a fuel-cell grade H₂ production (a PEMFC, i.e., 0.2 ppm of CO and 2 ppm of CO₂ in ISO 14687:2019)

Reviewer #2: (Please note that reviewer #2 did not provide any comments, but in a direct message to the editor mentioned that the manuscript was acceptable and asked the following questions about Figure 3d (breakthrough experiment):

The results shown do not reach the feed concentration (Fig. 3d). Why? Is it possible that reaction is occurring together with adsorption? If so how do you measure adsorption capacity

Response: We appreciate your careful review of our work.

As emphasized in the Introduction, the purpose of this study is the removal of trace CO in H₂ produced from various H₂ sources. From the viewpoint of applications, the applicable capacity (working capacity) is terminated when 5 ppm CO is detected in the breakthrough test, (Fig. 3(d)), instead of a full breakthrough test, because of the limitation of a fuel-cell grade H₂ (ISO 14687:2019). Nonetheless, to give more information to expected readers on how to approach the breakthrough curve after C/C₀ = 0.5, we presented the extended breakthrough curve (Fig. 3(d)).

In the study, the thermodynamic CO sorption capacities at various CO concentrations were measured using a separated system (a volumetric device) and presented by the adsorption isotherms (Fig. 3(a)). Adsorption amounts through breakthrough experiments are generally affected by thermodynamic adsorption capacities, heats of adsorption, and mass transfer resistances (film mass transfer, pore diffusion, surface diffusion, etc). Therefore, the adsorption capacity, evaluated by a breakthrough experiment, is considered a dynamic working capacity at an applied condition, and the capacity is generally smaller than the thermodynamic capacity. However, since the adsorption dynamics are out of scope, it was not fully discussed in the study.

To address two concerns of the reviewer's comment (1. The breakthrough curve will reach the feed CO concentration, 2. The potential of CO to CO₂ conversion during the breakthrough test), we have adapted the experimental setup to detect CO₂ accurately in the effluent stream during the CO breakthrough test. We configured a new analysis system consisting of a mass spectrometer along with a GC equipped with an FID and a methanizer, which enables CO₂ detection at ppb levels. Then, a new breakthrough test was conducted at 25°C until the breakthrough curve reached the feed concentration (50 ppm CO in H₂ balance).

In the result, any CO₂ was not detected in the effluent gas through the breakthrough

test, substantiating that MgCuCeO_x operates as a CO_2 -free CO sorbent and not as a catalyst for CO oxidation at 25°C . Therefore, we confidently confirmed that the proposed MgCuCeO_x sorbent could be used to simultaneously comply with the CO concentration criterion specified by ISO 14687:2019 without the issue of CO_2 generation during a CO removal step. We have revised the manuscript to present the results clearly with the addition of the new experimental breakthrough test as the supplementary data. Additionally, to present the goal of this study more clearly, the target breakthrough time ($C/C_0=0.01$: 50 ppm CO) was specified in the breakthrough result (Fig 3(d))

To describe the results clearly, the following corrections with additional experimental result were made in the revised manuscript.

Page 10, lines 202–210 (Revised)

The breakthrough time was determined at $C/C_0=0.01$ (0.5 ppm CO), marking the red dotted line in the figure. Given the purified H_2 mix from 0 minutes to the breakthrough time, its overall CO content is less than 0.2 ppm, and the sorbent should be regenerated after the breakthrough time. The detection times of 0.5 ppm CO in AC and AC- Cu^+ beds were specified in the insert figure, with the time for the $\text{Mg}_{13}\text{CuCeO}_x$ bed marked in green. The dynamic sorption capacity was calculated as the sorption amount up to the saturation time at $C=C_0$ ($S_{ads}(\text{ST})$) or a long-term duration of 5750 min ($S_{ads}(5750)$), thereby providing comparative data for understanding sorption kinetic effects in practical applicability.

Page 11, lines 216–223 (Revised)

As presented in Fig. 3d, $\text{Mg}_{13}\text{CuCeO}_x$ exhibited the highest performance with a breakthrough time of 1372 min compared to 3 min for AC and 44 min for AC- Cu^+ , indicating 31 times longer in terms of the breakthrough time. In addition, $S_{ads}(5750)$ of 0.209 mmol g^{-1} was also approximately 30 times higher than $S_{ads}(\text{ST})$ of AC- Cu^+ . $S_{ads}(5750)$ for $\text{Mg}_{13}\text{CuCeO}_x$ was 75% of the equilibrium sorption amount, although the contact time between the sample and CO was limited during dynamic operation. Given that $S_{ads}(\text{ST})$ of AC- Cu^+ was only 39% of the equilibrium capacity, $\text{Mg}_{13}\text{CuCeO}_x$ demonstrated a relatively rapid sorption rate during the initial breakthrough period.

Fig. 3d (Revised)

Fig. 3. (d) CO breakthrough curves at 25°C under a flow of 50 ppm CO and H₂ balance (GHSV = 935 h⁻¹).

Page 12, lines 234–249 (Added)

The CO species chemisorbed onto the sorbent surface were mainly desorbed as CO₂ at >100°C, with a small quantity of CO released between 100°C and 400°C. It indicated that the strongly chemisorbed CO begins to be desorbed in the form of CO₂ at elevated temperatures, instead of ambient temperatures. This phenomenon was corroborated by an additional breakthrough test (Fig. S5). At 25°C, any CO₂ molecules were not observed until the breakthrough curve reached the feed concentration, further emphasizing that CO was being chemisorbed without concurrent emission of CO₂. Furthermore, the absence of CO₂ at ppb levels during the breakthrough tests indicated that the MgCuCeO_x sorbent can satisfy the fuel-cell grade hydrogen quality with a successful trace CO and without CO₂ generation during a H₂ purification step. Consequently, the MgCuCeO_x sorbent functioned as a CO₂-free CO sorbent, rather than a catalyst for CO conversion, during CO removal step at ambient temperatures.

Trace H₂ was detected at around 330°C, which was associated with the interaction of CO with the OH groups on the sorbent surface (Fig. 2a). CO₂ peaks were observed at high temperatures (200, 330, and 580°C), representing a similar trend to that observed for the α, β, and γ peaks of Mg₁₃CuCeO_x in the TPR analysis (Fig. 2c).

Fig. S1 (Revised)

Supplementary data Page 8 (Added)

Fig. S1. Schematic diagram of breakthrough experimental apparatus.

In the long-term breakthrough experiment presented in Fig. 3d, we employed a monitoring approach illustrated by the blue line in Fig. S1, which exclusively focused on CO detection via infrared (IR) spectroscopy. Conversely, the breakthrough experiment presented in Fig. S5 was designed for the simultaneous detection of CO and CO₂ at ppb levels until the breakthrough curve reached the feed concentration. For this full breakthrough curve experiment, we used a more complex analysis setup that included a gas chromatograph (GC) equipped with a flame ionization detector (FID) and a methanizer, along with a mass spectrometer (MS). This experimental methodology is represented by the red line in Fig. S1.

Fig. S5 (Added)

Fig. S5. CO breakthrough curve at 25°C under a flow of 50 ppm CO and H₂ balance (GHSV = 4185 h⁻¹).

The breakthrough test was conducted under a harsher flow condition than the long-term breakthrough experiment displayed in Fig. 3d. The analysis system showed a noise level signal from the feed gas because of the CO₂ impurity in the feed gas. Then, through the breakthrough experiment, the noise level CO₂ signal did not change. The results revealed no detection of CO₂ even at CO₂ at ppb levels, and the breakthrough curve could reach 50 ppm of CO (the feed concentration). Considering the analysis system, which could detect CO₂ at ppb level, the MgCuCeO_x sorbent satisfied the removal of trace CO in H₂ for a fuel-cell grade H₂ production (a PEMFC, i.e., 0.2 ppm of CO and 2 ppm of CO₂ in ISO 14687:2019)

Reviewer #3: *In this study, bead-structured Mg, Cu, and Ce-based sorbent ($Mg_{13}CuCeO_x$) was synthesized for the removal of CO in hydrogen rich syngas for the purpose of using as fuel for hydrogen fuel cell. This is a very important step for the implementation of hydrogen fuel cell. It has the potential to be published in NC, but the current version is still far from my expectation of an NC paper. Some important points should be addressed in the next revision.*

Response: We appreciate the reviewer's detailed comments. We have revised the manuscript to address your concerns by providing additional experimental figures and explanations to clarify the conclusions. Since some comments resulted from our brief explanations to follow the journal's style and length limitation, we revised them with more explanatory details and references, and adjusted our phrasing to ensure description clarity and eliminate potential confusion. We hope that the revised version better meets the expectations of the *Nature Communications* journal.

Major comments:

Comment 1) *"Furthermore, powder-type sorbents have been reported to have a high CO sorption equilibrium at certain CO concentrations" I do not agree with this statement. The CO sorption equilibrium is a thermodynamic concept, so it should not be affected by the physical shape. The authors need to provide more explanations on how physical shape affect the equilibrium.*

Response: Thank you for your comment. We fully agree with the comment, which can lead to misunderstanding the effect of sorbent shape on sorption equilibrium, but the confusion might result from the terminologies.

According to our previous review paper for previously reported CO sorbents [ref. 16], the reported sorbents can be categorized by: powder and pellet. As the reviewer pointed out, the CO sorption equilibrium itself is a thermodynamic concept. However, during the pelletizing process to apply the powder sorbent for industrial processes, the specific surface area is reduced due to binders and/or pore clogging, which implies a decrease in the number of adsorption sites per unit weight and adsorption performance (Montazerolghaem et al., 2017; Rezaei et al., 2015).

Additionally, the reports suggested that the adsorption performance can be influenced by the presence, type, and content of binders added during the pelletizing process.

Since our intention was not to express the effect of the physical shape of a sorbent, the description was revised to avoid any confusion for expected readers.

Page 4, lines 66–71 (Revised)

Notably, while previous studies on powder sorbents have demonstrated a high CO sorption performance at 100–200 kPa of CO¹⁶, a research gap exists regarding CO sorption capacity at ppm levels. Simultaneously, investigations into applying them in a stable pellet form for practical process applications are also lacking. The pelletization of powder sorbents often results in reduced sorption capacity and stability due to a decrease in surface area and porosity in pellets¹⁸.

Page 10, lines 193–198 (Revised)

Numerous sorbents had reported considerable sorption performance, but powder sorbents often pose problems in packed or fluidized bed applications due to their associated high-pressure drop and potential for process contamination¹⁶. Thus, given these factors, a pellet Cu-impregnated activated carbon (AC-Cu⁺), with outstanding performance among the reported CO sorbents³⁷, served as a control group, and the performance evaluation incorporated both static and dynamic behavior.

Page 17, lines 347–350 (Revised)

Mg₁₃CuCeO_x exhibited superior CO sorption at both equilibrium and under dynamic conditions at ultra-low pressures below 10 Pa (≈100 ppm CO at a total pressure of 100 kPa), a performance that surpassed that of previously reported pellet sorbents.

Comment 3) "X-ray photoelectron spectroscopy (XPS) and N₂O chemisorption analysis were utilized to investigate the Cu⁺ surface area (denoted as S_{Cu⁺})" I think it is N₂ instead of N₂O. Another point is why Cu⁺ surface area is measured. Is not it the total surface area of the material? Does Cu⁺ have surface area?

Response: Thank you for the comments. The physical properties, including the total surface area were measured by N₂ isotherms at 77 K (BET surface area in Table 1), as commented by the reviewer. On the other hand, N₂O chemisorption was used to analyze the surface area of active sites in catalysts, which is a typical method using many previous studies, including ours (Jin et al., 2021; Y.-J. Liu et al., 2023). Since N₂O can be absorbed only at the surface of a specific metal at 50°C, the total Cu surface area could be evaluated by the N₂O chemisorption. Additionally, since Cu⁺ is the active site for the sorption of CO on MgCuCeO_x, we evaluated the surface area of Cu⁺ by multiplying the Cu⁺ ratio on the surface (obtained from XPS analysis) and total Cu surface area.

To clarify our intentions and results, the description and equation were modified as follows:

Page 6, line 97–99 (Added)

The physical properties of samples including BET surface areas were determined by the N₂ isotherms at 77 K. Since the total Cu surface area could be evaluated by the N₂O chemisorption, X-ray photoelectron spectroscopy (XPS) and N₂O chemisorption analysis were utilized to investigate the Cu⁺ surface area (denoted as S_{Cu⁺}).

Page 9, line 165 (Revised)

The influence of the Mg content on S_{Cu⁺} (Eqs. S3) and the chemical states of Cu and Ce species was investigated by combining XPS and N₂O chemisorption analysis (Figs. 2d, e, and Table 2).

Supplementary data Page 4–5 (Revised)

Cu dispersion and the specific Cu surface area (S_{Cu}) were determined using selective N_2O chemisorption experiments conducted at $50^\circ C$ following a well-documented methodology^{4,5}. Additionally, since Cu^+ is the active site for the sorption of CO on $MgCuCeO_x$, the surface area of Cu^+ of the as-prepared sorbent (S_{Cu^+}) was estimated by applying the surface Cu^+ ratio to S_{Cu} . Initially, the samples were reduced under a flow of 10% H_2/Ar at $400^\circ C$ for 1 h, and the H_2 consumption was measured by integrating the peak area. The chemisorbent bed was then cooled to $50^\circ C$ and flushed with He. The cooled chemisorbent was exposed to 10% N_2O/He gas for 30 min before being returned to room temperature and purged with He. An additional H_2 -TPR cycle was performed and the corresponding H_2 consumption was assessed. Cu dispersion (D) was calculated based on the H_2 consumption after the total oxidation of the catalyst in the first H_2 -TPR run (X) and subsequent oxidation of surface Cu atoms in the second H_2 -TPR run (Y). Cu dispersion and S_{Cu} were determined using Eqs. (S1) and (S2) respectively:

$$D = 2 \times Y/X \quad (S1)$$

$$S_{Cu} = (D \times N_{av} \times W_{Cu}) / (A_{Cu} \times 1.4 \times 10^{19}) \quad (S2)$$

where N_{av} is the Avogadro constant ($6.02 \times 10^{23} \text{ mol}^{-1}$), W_{Cu} indicates the Cu metal weight content determined using ICP-OES analysis, and A_{Cu} is the atomic weight of Cu ($63.546 \text{ g mol}^{-1}$). The ratio of Cu^+ to Cu^{2+} (M), obtained from the XPS analysis, was then used to calculate S_{Cu^+} by multiplying S_{Cu} by M (Eqs. (3)).

$$S_{Cu^+} = S_{Cu} \times M \quad (S3)$$

Comment 4) The experiment was conducted in 50ppm CO environment. However, most syngas has 10-50% CO. I would like to see the authors' material performance for high CO concentration. The integration of area below CO in Fig. 3d indicated it has pretty large CO capacity. So it might be possible to apply the material in high CO concentration gas.

Comment 7) The reason to choose Mg is not clearly demonstrated. Please elaborate the reason why Mg should be used in this study.

Response: We appreciate the reviewer's suggestion regarding the possibility of expanding the application of our material.

As mentioned by the reviewer, H₂ effluent streams from various resources (such as SMR, BFG, LDF, COG, IGCC, etc) contain a certain range of CO (5 to 50%). Generally, almost industrial separation processes focus on production 99.99+% H₂. Although the produced H₂ gases can be applicable to current chemical processes, it is hard to apply them to fuel-cell because they contain a certain amount of CO in H₂ (addition of Table S1). On the other hand, to satisfy the fuel-cell H₂ grade in present processes, the H₂ cost increased steeply due to the reduction of H₂ recovery and productivity.

The goal of this study is to remove trace CO (ppm levels) in produced H₂ from various sources for fuel-cell grade H₂ production, which is different from sorbents for bulk CO removal (percent levels) as reported in the review article by our group (Ko et al., 2022: ref [16]). Like the difference between bulk phase and dilute solution in thermodynamics, material development and applications for very gas adsorption should be separately studied. Therefore, the reported sorbents, functioned by quasi-chemisorption with CO, do not work effectively at low pressures (the range of trace CO concentrations), as demonstrated in Fig. S4. On the other hand, previously reported sorbents can have advantages for bulk separation of CO (high concentrations) because the CO isotherm showed the crossover at 1.7 kPa CO in the same figure.

Enhancing the oxygen mobility of the material surface could strengthen the interaction with CO (Chen et al., 2023: ref [24]). In catalytic reaction studies, it was suggested that introducing Mg into Cu could increase surface oxygen mobility, and the conversion efficiency was highly enhanced (Jin et al., 2021: ref. [23, 35]). In addition, Mg source is cheap and

abundant compared to other materials. Therefore, Mg was selected as a candidate material in the study.

To describe the purpose of this study more clearly, the following corrections were made by addition explanations and references, including H₂ products with CO in ppm levels (refer to the added Table S1) and the structural and electrochemical advantages of Mg.

Page 5, lines 84–87 (Added)

It has been reported that incorporating magnesium into a Cu–Ce system promotes redox mechanisms, thus enhancing the water–gas shift reaction ⁵. In particular, previous studies suggested that the introduction of Mg into Cu increases oxygen mobility ²³, which can intensify the interaction between CO and the material surface ²⁴.

Page 4, lines 65–74 (Revised)

To overcome these limitations, it is essential to develop sorbents with a robust CO sorption capacity (i.e., within the ppm range) that do not generate CO₂ and thus avoid the need for additional separation processes. However, the majority of CO sorbent research to date has concentrated on enhancing CO sorption within a pressure range of 100–200 kPa, which is more useful for CO production than for removal ^{16,17}. Additionally, H₂ products from H₂ production or purification processes contain a few percent or ppm of CO (Table S1). Notably, while existing studies on powder sorbents have demonstrated a high CO sorption performance at 100–200 kPa of CO ¹⁶, a research gap exists regarding CO sorption capacity at ppm levels. Simultaneously, investigations into applying them in a stable pellet form for practical process applications are also lacking. The pelletization of powder sorbents often results in reduced sorption capacity due to a decrease in surface area and porosity in pellets ¹⁸. Given these considerations, the development of feasible materials that not only offer high sorption capacity at CO ppm levels, but also maintain process stability in pelletized form, is necessary for practical applications.

Table S1 (Added)**Table S1.** H₂ purity and CO concentration in effluent gas from H₂ production and purification processes

Process	Feed	H ₂ purity [%]	CO concentration in produced H ₂	References
SMR + WGS	NG	75–80	0.1–4 %	^{7–10}
PSA	SMR syngas	99.959	-	¹⁰
PSA	SMR syngas	99.884–99.987	0.78–27.05 ppm	¹¹
PSA	Coal off-gas	99.96–99.99	1.1–6.7 ppm	¹²
VSA	SMR syngas	99.933–99.991	0.18–15.55 ppm	¹¹
TSA	SMR syngas	99.903–99.992	0.17–33.42 ppm	¹¹
VPSA	SMR off-gas	99.981	63 ppm	¹³

Comment 5) Fig. 1a showed a bead. I would like to see a larger range of the picture. Is everywhere the material ball shape? Or only some of them are in ball shape?

Response: In response to the reviewer's queries about the bulk form of the sorbent, the following photograph of the sorbent was added to the supplementary information.

Page 7, line 127 (Added)

The scanning electron microscopy (SEM) image in Figs. 1a, b shows that $\text{Mg}_{13}\text{CuCeO}_x$ has a spherical bead morphology with a smooth surface, and the photograph of the bulk sorbents is presented in Fig. S3.

Figure S3 (Added)

Fig. S3. Photograph of MgCuCeO_x beads.

Comment 8) The authors criticized CO-PROX technology due to the generation of CO₂. What if the system loads some CO₂ sorbents? Is the problem possibly to be resolved?

Response: Thank you for your insightful comment. While it's certainly true that CO₂ adsorbents can synergistically operate with PROX catalysts, it's important to note that optimal operation for most current PROX research falls within the temperature range of 80–150°C (H. Liu et al., 2023), not at ambient temperatures (25°C in the study). It indicates the need for thermal energy for the reaction. In addition, due to CO to CO₂ conversion, adsorbents with strong adsorption affinity of CO₂ are required for trace CO₂ removal. However, it leads to additional high energy consumption (thermal or vacuum energy) to regenerate adsorbents due to the strong adsorption affinity. Furthermore, the removal of excess oxygen supplied in CO-PROX is an additional issue in fuel-cell grade H₂ production. On the other hand, the MgCuCeO_x system developed in the study uniquely removed trace CO directly at room temperature, but it also needed thermal energy for the sorbent regeneration (using air at 120°C). Therefore, the CO-PROX is more energy-intensive and complex compared to the developed sorbent system.

We do not intentionally want to describe certain disadvantages of CO-PROX for trace CO removal in detail compared to the MgCuCeO_x system because CO-PROX is also one of the important research areas. Therefore, we simply cited the reference to describe the issues. Instead of criticizing certain limitations of CO-PROX, we modified the description to emphasize the unique objectives and applications of our study.

Page 4, lines 57–60 (Added)

and (2) it produces CO₂, meaning that it fails to meet ISO standards requiring CO₂ levels lower than 2 ppm¹⁵. Therefore, the CO-PROX process needs reaction energy, and additional separation unit for the removal of CO₂ and excess O₂ with additional regeneration energy. As a result, compliance with ISO standards using current H₂ purification methods is difficult due to the requirement of complex and energy-intensive processes.

Page 18, lines 369–371 (Added)

Consequently, the presence of Mg contributed to improving the structural and chemical properties of the sorbent, thus enhancing its sorption capacity and cyclic stability. Therefore, MgCuCeO_x represents a promising, effective, and stable sorbent for ultra-low levels of CO, one that does not discharge CO₂ during the removal process. Since MgCuCeO_x proved a high removal potential of trace CO in H₂ and was readily regenerated at 120°C under an air flow, simple systems using MgCuCeO_x can cost-effectively remove trace CO.

Minor comments:

Comment 2) *The above sentence should have a comma instead of a full stop at the end.*

Comment 6) *The caption of Fig. 2 has typo. Fig. 2e's caption is missing.*

Comment 9) *PROX's full name is not provided in this article.*

Response: We acknowledge the typos, as noted in comments 2, 6, and 9, and we thank the reviewer for the careful attention. Based on the reviewer's comments, the manuscript was revised to correct the typos and clarify terms.

Page 3, line 48 (Revised)

Numerous H₂ purification methods have been proposed to comply with these stringent H₂ purity standards, including membrane separation^{5,9}, pressure (vacuum) swing adsorption (P(V)SA)^{10,11}, and preferential oxidation of CO (PROX)^{12,13}.

Fig. 2 title (Revised)

Fig. 2. Physicochemical analysis of the as-prepared sorbent samples: (a) DRIFTS spectra (b) XRD patterns, (c) H₂-TPR results, and XPS profiles : (d) Cu2p and (e) Ce3d spectra.

References

Chen, J., Xiong, S., Liu, Haiyan, Shi, J., Mi, J., Liu, Hao, Gong, Z., Oliviero, L., Maugé, F., Li, J., 2023. Reverse oxygen spillover triggered by CO adsorption on Sn-doped Pt/TiO₂ for low-temperature CO oxidation. *Nat Commun* 14, 3477. <https://doi.org/10.1038/s41467-023-39226-6>

Jin, S., Byun, H., Lee, C.-H., 2021. Enhanced oxygen mobility of nonreducible MgO-supported Cu catalyst by defect engineering for improving the water-gas shift reaction. *Journal of Catalysis* 400, 195–211. <https://doi.org/10.1016/j.jcat.2021.05.030>

Ko, K.-J., Kim, H., Cho, Y.-H., Lee, H., Kim, K.-M., Lee, C.-H., 2022. Overview of carbon monoxide adsorption performance of pristine and modified adsorbents. *J. Chem. Eng. Data* 67, 1599–1616. <https://doi.org/10.1021/acs.jced.1c00903>

Liu, H., Li, D., Guo, J., Li, Y., Liu, A., Bai, Y., He, D., 2023. Recent advances on catalysts for preferential oxidation of CO. *Nano Res.* 16, 4399–4410. <https://doi.org/10.1007/s12274-022-5182-9>

Liu, Y.-J., Kang, H.-F., Hou, X.-N., Qing, S.-J., Zhang, L., Gao, Z.-X., Xiang, H.-W., 2023. Sustained release catalysis: Dynamic copper releasing from stoichiometric spinel CuAl₂O₄ during methanol steam reforming. *Applied Catalysis B: Environmental* 323, 122043. <https://doi.org/10.1016/j.apcatb.2022.122043>

Montazerolghaem, M., Aghamiri, S.F., Talaie, M.R., Tangestaninejad, S., 2017. A comparative investigation of CO₂ adsorption on powder and pellet forms of MIL-101. *Journal of the Taiwan Institute of Chemical Engineers* 72, 45–52. <https://doi.org/10.1016/j.jtice.2016.12.037>

Rezaei, F., Sakwa-Novak, M.A., Bali, S., Duncanson, D.M., Jones, C.W., 2015. Shaping amine-based solid CO₂ adsorbents: Effects of pelletization pressure on the physical and chemical properties. *Microporous and Mesoporous Materials* 204, 34–42. <https://doi.org/10.1016/j.micromeso.2014.10.047>

Reviewer #2 (Remarks to the Author):

The revised version of the manuscript is acceptable and the reply to my comment is adequate.

Reviewer #3 (Remarks to the Author):

The authors have addressed my comments in a detailed manner. I reckon the manuscript can be accepted for now.